# Functionalized calcium carbonate microparticles in ethyl cellulose films: A vehicle for sustained amoxicillin release for medical applications

Petru Niga[1]*, Simone Sala[1], Jenny Rissler[1], Lina Nyström[1], Anna Fureby[1,2],
Ulla Elofsson[1], Joachim Schoelkopf[3], Roger Roth[3], Patrick Gane[3,4,5]

1 Chemical Process and Pharmaceutical Development, RISE – Research Institute of Sweden, Stockholm, Sweden, 2 Division of Food and Pharma, Lund University, Lund, Sweden, 3 Omya International AG, Egerkingen, Switzerland, 4 Department of Bioproducts and Biosystems, Aalto University, Helsinki, Finland, 5 Faculty of Technology and Metallurgy, University of Belgrade, Belgrade, Serbia

* petru.niga@ri.se

## Abstract

The continuous quest for materials capable of providing sustained release of antimicrobial drugs is particularly important for indwelling medical applications. In this study, we utilized amoxicillin as a model active pharmaceutical ingredient (API) to investigate the feasibility of using porous media – specifically, functionalized calcium carbonate (FCC) microparticles – as a primary drug carrier embedded within an ethyl cellulose (EC) polymer film. Our main objective was to prolong and sustain the release of the API. The fabrication process of the microparticle containing film involved two key steps: loading the model API into the FCC particles and then embedding these loaded particles into the polymeric film. Amoxicillin was loaded into the FCC particles using a solvent evaporation method. Detailed characterization through Scanning Electron Microscopy (SEM), lab- and synchrotron-based XRD revealed that amoxicillin precipitated both inside and on the surface of the FCC particles, predominantly in an amorphous form. Additionally, ultraviolet-visible (UV-vis) spectroscopic data demonstrated an increased release rate from the porous FCC compared to direct dissolution of pure amoxicillin powder. Embedding amoxicillin preloaded porous FCC particles in the EC film led to a more rational sustained release compared with powder amoxicillin embedded directly in the film, advantageously delivering the same amount of amoxicillin over a longer period; a result that may be relevant for indwelling medical devices such as urinary catheters, vascular access devices or wound drains.

## Introduction

Sustained drug release offers several advantages over conventional drug delivery systems. In conventional delivery, the blood plasma concentration of a drug typically

**Data availability statement:** All relevant data are within the manuscript.

**Funding:** This work was supported by Omya International AG and RISE Research Institute of Sweden. We acknowledge MAX IV Laboratory for time on Beamline NanoMAX under Proposal 20230043. Research conducted at MAX IV, a Swedish national user facility, is supported by the Swedish Research council under contract 2018-07152, the Swedish Governmental Agency for Innovation Systems under contract 2018-04969, and Formas under contract 2019-02496. There was no additional external funding received for this study. The funders (Omya International AG and RISE Research Institutes of Sweden) participated in study design, data collection and analysis, decision to publish, and preparation of the manuscript.

**Competing interests:** The authors have declared that no competing interests exist.

increases abruptly after administration, peaking above the maximum desired level, which may lead to adverse effects. Subsequently, the concentration drops below the minimum effective level, rendering the drug ineffective, and increasing the risk of developing bacterial resistance [1]. In contrast, sustained drug delivery requires a specific controlled release rate dependent on the loading level of active ingredient, and the dynamic of release from the porous medium. This approach is particularly beneficial for patients with low compliance [2].

Amoxicillin, a β-lactam penicillin derivative, discovered in 1958 by Doyle, Nayler and Smith [3], that can withstand the acidic environment of the stomach, is widely used to treat various bacterial infections and is one of the most commonly prescribed antibiotics in primary care settings [4]. However, the extensive use of antibiotics has led to the rise of antimicrobial resistance, which is one of the most pressing challenges in modern medicine [2]. To combat this, researchers are focusing on discovering new antibiotics and developing formulations that limit antimicrobial resistance. Sustained drug release systems are of particular interest because they can maintain drug levels above the minimum inhibitory concentration (MIC), without increasing the total amount of drug released, thereby reducing the likelihood of resistance development.

Sustained release formulations of amoxicillin for oral delivery have been extensively studied. For instance, the mucoadhesive sustained release of amoxicillin from Eudragit RS100 microspheres demonstrated that the release profile depended on both the mucosa and the swelling ability of the polymer, sustained release for approximately 12 h [5]. Additionally, a 24 h sustained release was achieved by loading amoxicillin into composite hydrogels made of poly(acrylamide) and starch, where release kinetics were sensitive to both temperature and pH [6]. Sustained release has also been achieved by coating amoxicillin particles with ethyl cellulose (EC), with additional chitosan or chitosan-cyclodextrin coatings further prolonging the release up to 24 h [7]. Various porous carriers such as organically modified MCM-41 (Mobil Composition of Matter No. 41) silica [8] and magnesium-doped hydroxyapatite (Mg-HA) [9] have been shown to modulate amoxicillin release through diffusion-controlled mechanisms and surface interactions, yet their release duration is limited to hours or a few days. Additionally, inorganic hollow silica particles were also used as drug carriers. This nanospheres system demonstrated a three-stage amoxicillin release in simulated body fluid for about 25 h [10]. However, such systems typically lack long-term release potential, limiting their utility in applications requiring sustained antimicrobial action over weeks or months.

For indwelling applications, such as catheters and implants, sustained release must be controlled over longer periods to maintain therapeutic efficacy. For example, hydroxyapatite nanoparticles loaded with amoxicillin and coated with polyvinyl alcohol or sodium alginate achieved a sustained release over 30 days, which was explored for treating bone infections [11]. Additionally, for root tooth infections, a formulation of amoxicillin-loaded Eudragit S100 microparticles mixed in reparative cement was investigated. These results showed that this formulation effectively delayed the drug release, specifically after 96 hours only 16% of the amoxicillin was released [12].

Currently, significant efforts are directed at developing new indwelling urinary catheters (IUCs) designed to prevent bacterial infections, a major healthcare and patient well-being concern. Despite advances in antifouling and antibacterial catheter coatings, translating these concepts into clinical practice remains challenging due to discrepancies in experimental evaluations. For a comprehensive review of materials and their effectiveness, readers are referred to Andersen and Flores-Mireles, 2020 [13–16].

Recent advances in nanocomposite-based antimicrobial systems have highlighted the potential of natural halloysite nanotubes (HNTs) as biocompatible carriers for sustained antibiotic release. For instance, HNTs have been successfully used in combination with polymers such as chitosan or polycaprolactone to deliver antibiotics like norfloxacin, erythromycin, and diclofenac with prolonged antibacterial effects against both Gram-positive and Gram-negative bacteria [17–19]. These systems provide inspiration for developing similarly robust and sustained-release antibacterial platforms applicable to infection-prone environments such as urinary catheters.

The terms *extended release* and *sustained release* are sometimes used interchangeably, but they carry different meanings. *Extended release* refers broadly to any system that delays the release of a drug compared to immediate-release forms. *Sustained release*, by contrast, specifically describes systems designed to maintain a consistent therapeutic level of the drug over an extended duration. Here, we use the term *sustained release* to reflect the controlled and prolonged delivery of amoxicillin from the ethyl cellulose film matrix.

In this study, we explore the feasibility of using a polymeric matrix containing porous particles loaded with amoxicillin as a model antibiotic to form a platform to exemplify the sustained release of an active pharmaceutical ingredient (API), for use in catheter applications. This application would strongly benefit from sustained release, which cannot be achieved by simply mixing the active ingredient into the polymer matrix (also confirmed in this study). The porous particles used in this study are functionalized calcium carbonate (FCC). They are composed of calcium carbonate surrounded by an intertwined layer of calcium phosphate platelets, offering enhanced specific surface area and high interconnected porosity [20–22].

Although amoxicillin sodium salt is readily water-soluble and does not typically exhibit dissolution issues related to hydrophobicity, the use of porous carriers remains highly relevant for applications requiring sustained therapeutic levels. In many cases, organic drugs tend to self-assemble into hydrophobic agglomerates at the air–liquid interface due to their molecular composition, which can hinder dissolution. By contrast, depositing a monomolecular layer onto a substrate can preclude this interfacial rearrangement, enabling rapid initial dissolution. As demonstrated in previous work on flavorings and nutraceuticals [21], this approach can be effective for otherwise hydrophobic compounds. The primary objective is not to improve dissolution but to retain a depot of amoxicillin in the porous matrix and support a continuous release above the minimum inhibitory concentration (MIC) to ensure persistent antibacterial activity over prolonged timeframes.

In the first step of this study, we investigated the loading and release profile of amoxicillin from the FCC particles. The state of the precipitated amoxicillin in the FCC was analyzed using SEM and lab- and synchrotron-based XRD. In the second step, the amoxicillin-loaded FCC particles were uniformly dispersed in ethyl cellulose (EC) and cast into a film. Additionally, the release profile of amoxicillin from this film in water was evaluated using UV-vis spectrometry.

## Materials and methods

### Materials

Amoxicillin was obtained from Sigma Aldrich with a purity higher than 99.9%. Absolute ethanol (≥99.8%), acetone (≥99.5%), methanol (≥99.8%), toluene (≥99.9%), and methyl ethyl ketone (MEK) (≥99.0%) were also sourced from Sigma Aldrich. The Milli-Q water (resistivity 18.2 MΩ·cm) was from a Milli-Q purification system (Millipore). Porous functionalized calcium carbonate (FCC) particles were supplied by Omya International AG, Switzerland. Three types of functionalized calcium carbonate (FCC) materials were used in this study: FCC fines, FCC granules, and the commercial Omyapharm FCC500-OG. These materials share identical chemical composition but differ in particle size, degree of structuration,

**Table 1. Comparative overview of FCC materials used in this study.**

| Material Type | Particle Size (μm) | SSA* (BET**, m²/g) | Bulk Density (g/cm³) | Morphology/ Structural Features | Experimental Use/ Rationale |
|---|---|---|---|---|---|
| FCC fines | 0–180 | 43.4 | Loose: 0.23; Tapped: 0.27 | Individual microparticles with high surface area and open porosity | Used to study the effect of smaller particle size and higher surface area on amoxicillin loading and release kinetics |
| FCC granules | 180–710 | 40.8 | Loose: 0.55; Tapped: 0.59 | Aggregates of fines forming larger granules; inter-particle voids introduce additional macroporosity | Used to assess how structuration of FCC particles into granules affects loading and release (mass transport in inter-particle pores). |
| Omyapharm (FCC 500-OG) | $d_{50}$=6.6 (median) | 53.0 | Apparent: 0.13 | Commercial FCC material – identical chemistry to fines/granules | FCC source for film preparation with EC; selected for commercial reproducibility and representative FCC family structure |

*) Specific Surface Area

**) Brunauer–Emmett–Teller (a surface area analysis method)

and bulk density. Their key physicochemical parameters and experimental roles are summarized in Table 1. The granular material has been shown earlier to accelerate advantageously pharmaceutical tablet disintegration in aqueous medium, for example, as illustrated in these references [23–25]. The context of exemplifying structuration of FCC in respect to release kinetic of amoxicillin is, therefore, undertaken here to establish if such structuration provides any release mechanistic advantages, stressing once again throughout that the particle structure and chemical nature of the particles remain intrinsically constant. Omyapharm FCC500-OG illustrates the current commercially available product from which the experimental fractions were similarly derived.

Particles were dried at 200 °C for 2 h before processing. Ethyl Cellulose N100 (Ethoxyl grade 48–49.5%) was generously provided by IMCD Group. Water was sourced from a MilliPore RiOs-8 system. Reference amorphous amoxicillin used in synchrotron experiments was prepared by solvent evaporation method in a rotary evaporator. Amoxicillin was loaded into porous particles at 15% and 30% w/w (weight percent, calculated as mass of amoxicillin relative to the total mass of the loaded sample). Ethyl cellulose (EC) and functionalized calcium carbonate (FCC) are recognized as biocompatible materials commonly used in pharmaceutical and biomedical applications.

## Particle loading

To load amoxicillin into the porous FCC particles, the amoxicillin must first be dissolved in a solvent with a low boiling point and associated high vapor pressure. This solution is then mixed with the FCC particles, as present in all structural forms used in the study, namely FCC fines, granules and commercial Omyapharm, and the solvent is allowed to evaporate. During evaporation, amoxicillin precipitates both within the pores and on the surface of the FCC particles. The FCC particle structure is known to be stable when exposed to organic solvents [26].

In this study, a 3 g/L solution of amoxicillin in an ethanol/acetone mixture was prepared. This solvent system was selected due to the low boiling points of both components, which allow efficient solvent removal under mild conditions. This solution was then added to a predefined amount of particles (for 30 w/w% and 15 w/w% loading – 7 g FCC and 8.5 g FCC, respectively, per 1 L solution) and the solvent was evaporated using a rotary evaporator (30 °C and 400−150 mbar). It is recognized that the distribution of amoxicillin in the case of the granulated structure will include loading between particles, and not only on the surface, and within the pores of the individual particles. Given the nature of the granule concept, material between the particles, to a first approximation, is likely to behave similarly to that associated with the exterior of the particles, though it should be recognized that access of liquid to the interior of a granule may be hindered to reduced permeability. Further optimization will be considered in the future.

## Film preparation and fabrication

Ethyl cellulose (EC) was selected as the film-forming polymer due to its excellent biocompatibility, film-forming ability, and widespread use in controlled drug release applications [27]. As a non-biodegradable yet biodurable cellulose derivative, EC provides a robust barrier for drug release, as its physicochemical properties, such as solubility and viscosity, can be tailored by adjusting the degree of etherification. These properties make EC a suitable matrix for long-term drug delivery systems.

Film preparation was carried out using organic solvent evaporation to avoid high-temperature curing, which could degrade the thermally sensitive amoxicillin [28]. To minimize the dissolution of the active ingredient into the film matrix, an organic solvent mix (toluene: ethanol) was selected that dissolves ethyl cellulose (EC) efficiently while offering minimal solubility for amoxicillin.

For film production, Omyapharm particles, the starting material for FCC fines and granules, (which retains the structural and chemical properties of FCC particles family) were loaded with 30 w/w% amoxicillin using a solvent evaporation process with a MEK/methanol mixture. For the polymer solution, 5 g of EC-N100 grade polymer was dissolved in 100 mL of an 80:20 w/w% toluene: ethanol mixture, resulting in a clear, viscous solution. Upon adding the amoxicillin-loaded Omyapharm particles, a white dispersion was obtained, as illustrated in Fig 1 A. This suspension did not show any signs of agglomeration or sedimentation for more than 1 h, which was sufficient to produce the films. The films were applied onto a low-density polyethylene (LDPE) substrate using a hand coater (Erichsen – Germany) with a 200 μm indenture. Note that the final dried film thickness was not measured; throughout this paper, we refer to film thickness in relation to the size of the edge indenture (e.g., 200 μm).

The maximum particle content (Omyapharm) within the total solid content used in the dispersion formulation was defined as the value at which the resulting film adhered well to the substrate over time without peeling. This maximum was found to be 10 w/w% of total solid content for all film thickness tested. Additionally, it was visually observed that the distribution of Omyapharm particles throughout the film were uniform at this concentration. Fig 1 shows 30 w/w% amoxicillin-loaded Omyapharm particles suspended in a solution of EC-N100 (Fig 1A), and the final film coated on the LDPE substrate (Fig 1B).

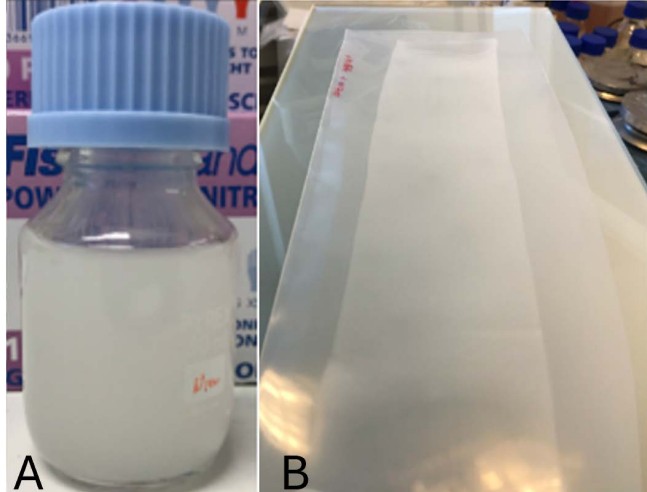

**Fig 1. A) 30 w/w% amoxicillin-loaded Omyapharm suspended in a solution of EC-N100 dissolved in toluene: ethanol mixture, and B) a film coated on a LDPE substrate.**

## Experimental setup for release studies

All release experiments were carried out in Milli-Q water at a starting pH of 6.99 and a temperature of 22 °C. For the release from particles, the weight of the samples was chosen to maintain sink conditions throughout the duration of the experiments. Particles and pure amoxicillin were placed in a beaker, containing 75 mL of Milli-Q water and stirred at 100 min⁻¹ (RPM – revolutions per minute). The dissolution media was circulated via a peristaltic pump into the UV-spectrometer and returned back into the beaker. Absorbance was measured at 272 nm. The concentration was calculated using a linear calibration curve (shown in Supporting Information) which was similar to previous literature records) [6].

For the experiments testing release from the film, 200 μm thick films containing 10 w/w% of particles loaded with amoxicillin were used. The measurements were carried out in 5 mL vials and samples (1 mL) were manually withdrawn at specified time intervals and returned after UV-vis measurement. The film samples were not agitated.

To adjust the surface area exposed to water, a special configuration of the film was employed, such that the film was perforated with a pin and a wire passed through the holes to keep the adjacent film roll apart and allow water to penetrate, as seen in Fig 2. The film surface area exposed to water in this way was calculated to be 5.4 cm²/mL. Therefore, drug-release experiments were performed under non-sink conditions (using a fixed volume of Milli-Q water without medium renewal), in order to mimic the limited fluid exchange surrounding indwelling medical devices such as catheters. A concise summary of key experimental parameters, including sink/non-sink conditions, release medium volume, temperature, agitation, and exposed surface area, is provided in S2 Table (Supporting Information) to facilitate reproducibility.

## Material characterization and techniques

**Loading Level and Release Studies.** The loading level and extent of amoxicillin release in water were determined using a thermogravimetric analyzer (TGA 2 Stare System, Mettler Toledo, Columbus, OH, USA) with a STARe System software. The system was heated from 25–800 °C at a rate of 20 °C/min. The amount of amoxicillin in the FCC particles was measured by analyzing the material's weight loss as it was heated. The rationale [29,30] for using TGA is as follows.

TGA measures weight changes in a sample as it is heated, allowing for the detection of components that volatilize at different temperatures. In our context:

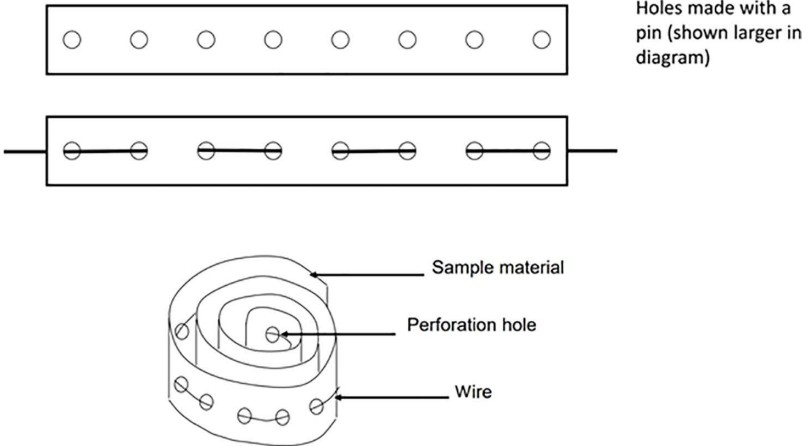

**Fig 2. Configuration of the FCC–ethyl cellulose film sample used for release testing.** The film was perforated with pin-sized holes to facilitate uniform contact with the release medium. A stainless-steel wire was threaded through the perforations to hold the film in a coiled geometry during immersion in the release beaker, ensuring stable positioning and full exposure to the aqueous medium.

1. Organic Material in Inorganic Matrix: Amoxicillin, an organic compound, decomposes at a lower temperature compared to the inorganic FCC matrix. This difference enables the quantification of amoxicillin content based on weight loss observed during TGA.

2. Post-Release Analysis: After the release study, the FCC particles were separated by centrifugation, and the collected FCC cake was analyzed using TGA. The weight loss corresponding to the decomposition of amoxicillin provided an estimate of the drug retained in the particles, thereby allowing us to infer the amount released into the aqueous medium.

### Imaging and spectroscopic analysis

Imaging was performed using a Quanta™ FEG 250 Scanning Electron Microscopy SEM (FEI Instruments, Hillsboro, OR., USA). The concentration of amoxicillin in water was measured using a Lambda™ 650 UV-vis spectrometer (PerkinElmer, Shelton, CT, USA) by recording absorbance at 272 nm, with concentration determined by interpolation from a calibration curve. It is important to note that FCC does not absorb at 272 nm, the wavelength specific to amoxicillin, and thus does not contribute to the UV-Vis signal in the release studies.

### Crystal structure

To investigate the crystal structure of amoxicillin, two X-ray diffraction (XRD) instruments were employed:

i. Lab Bench X-Ray Diffractometer: An X'Pert PRO diffractometer (Malvern Panalytical, Malvern, UK) was used under the following conditions: temperature at 26.5 °C, a 1 mm thick compacted powder sample, Bragg reflection geometry, spinning sample holder, and a zero-background holder with a Si wafer. Use of the Cu anode's Kα emission line resulted in an incident photon energy of 8.05 keV. The resulting diffractograms were indexed and fits were refined using the standard software HighScore (Malvern Panalytical).

ii. Synchrotron Hard X-Ray Nanoprobe Beamline: Measurements were conducted at the NanoMAX beamline at MAX IV, Sweden [31,32]. The incident photon energy was set to 15 keV and the XRD signal was collected using the Pilatus3 X 1M detector (Dectris, Baden, Switzerland), with 981 × 1043 pixels, each measuring 172 µm. To achieve a higher photon flux ($1.1 \times 10^{10}$ photons/s), the beamline's slits were slightly opened, resulting in a partial loss of coherence and a focused X-ray beam of 80 nm × 80 nm. Two-dimensional (2D) XRD measurements were carried out by scanning the sample through the nano-focused X-ray beam with a step size of 80 nm, matching the focal spot size. This generated 2D maps with a lateral resolution of 80 nm and for which a diffraction pattern is available at every pixel. Simultaneous detection of the characteristic X-ray fluorescence (XRF) emission from sulfur was carried out using the beamline's single-element silicon drift detector (SDD) (RaySpec, High Wycombe, UK). As amoxicillin was the only compound containing sulfur within the investigated samples, sulfur distribution was used as proxy for amoxicillin distribution. For the experiments, single particles were deposited by electrostatic precipitation from air on thin $Si_3N_4$ membranes (1 µm thick). We note that the nanoscale XRD measurements provide qualitative structural contrast only, as the limited probed area and low scattering intensity do not permit quantitative refinement of crystalline parameters.

## Results and discussion

### Drug-load in porous particles

Amoxicillin, the model API, was loaded into the porous particles using the solvent evaporation method. Loading levels were quantified via TGA, as shown in Table 2. The measured loading levels were slightly lower than the calculated values, likely due to amoxicillin precipitating on the walls of the rotary evaporator during solvent evaporation.

**Table 2. Calculated and measured amoxicillin loading values in porous particles by TGA.**

| Method | Amoxicillin concentration in | | | |
|---|---|---|---|---|
| | FCC fines/ w/w% | | FCC granules/ w/w% | |
| Calculated | 15 | 30 | 15 | 30 |
| Measured | 13 | 22.5 | 12 | 27 |

As shown in Fig 3, the release profile exhibited a two-stage pattern: an initial rapid release within the first 2–3 min, followed by a slower release over the next 30 min, approaching (but not reaching) a plateau. After 30 min, the 30 w/w% and 15 w/w% amoxicillin-loaded FCC fines reached concentrations of 99 w/w% and 94 w/w% of the total amount, respectively. After 24 h, all of the loaded amoxicillin was released into the solution, as confirmed by TGA (see Supporting Information). The release behavior from FCC granules followed a similar trend (also recorded in the Supporting Information). Even though in both cases the amoxicillin was fully released in water, the release curves of the two samples show different trends. That may be because the amoxicillin on the outer part of the particles is released more easily (which is assumed to be more in the 30 w/w% case) while amoxicillin adsorbed in the inner part, close to the pore wall surface, is released more slowly.

To assess the benefits of loading amoxicillin into porous media (specifically FCC granules), a comparative experiment was conducted. In this experiment, the release profile of 30w/w% amoxicillin-loaded FCC granules was compared to that of amoxicillin powder (as purchased) under identical conditions. For both forms, the quantity of amoxicillin added to the beaker was calculated to be below the solubility limit (1 mg/mL) if fully released. As illustrated in Fig 4 during the initial rapid release phase (from 3 to 3.6 min), the slope of the fitted release curve was slightly higher for the amoxicillin released from FCC granules (1.22 w/w%/min) compared to the powder form (1.083 w/w%/min). Additionally, while all the amoxicillin loaded in FCC granules was released within the experiment's duration, only 95 w/w% of the as-purchased powder form dissolved. These findings indicate that amoxicillin loaded in FCC granules not only releases more quickly but also more completely in water compared to the powder form. This suggests an enhanced dissolution rate, which is not typically observed for crystalline amoxicillin.

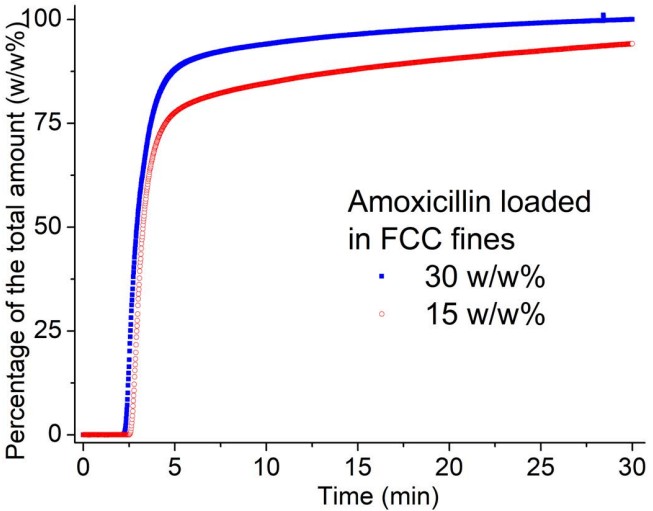

**Fig 3. The amoxicillin release curves in water from 15 w/w% (red circles) and 30 w/w% (blue squares) amoxicillin loaded FCC fines.**

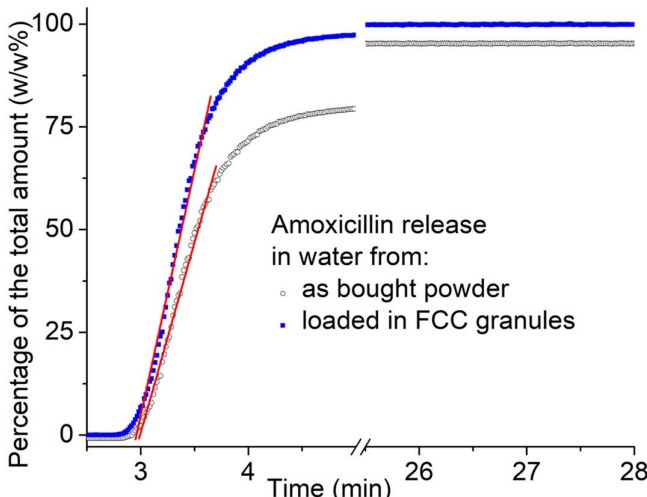

**Fig 4. Amoxicillin release in water from 30w/w% amoxicillin-loaded FCC granules (blue squares) compared to pure amoxicillin powder (black circles).** The red line is the fitting curve from 3 to 3.6 min. Data shown corresponds to one representative experiment.

These release curves, in Fig 3 and Fig 4, represent single representative experiments; although the measurements were repeated, variations in water volume and hydrodynamic conditions prevented calculation of meaningful average data.

## XRD experiments

To understand the underlying mechanisms influencing amoxicillin release, the structural characteristics of the loaded amoxicillin were examined using both lab bench and synchrotron-based XRD (2D-mapping).

## Lab bench XRD experiments

For the XRD analysis, three samples were investigated: (i) amoxicillin powder, (ii) amoxicillin loaded into FCC fines, and (iii) FCC fines as a reference. The crystal cell parameters of the amoxicillin powder were determined as follows: unit cell volume = 1781.47 Å³, with parameters a = 14.76 Å, b = 20.85 Å, c = 5.79 Å, and β = 92.53°. These values – albeit obtained from powder diffraction data – closely align with the orthorhombic single crystals reported in the literature [33–35].

The diffraction pattern of amoxicillin powder, shown in Fig 5, exhibits numerous peaks in the $q$ = 7–46 nm⁻¹ range, indicating a highly crystalline phase. The recorded intensities, reaching up to 400k counts, exceed those reported in previous studies due to the extended data collection time employed in this work [7]. In contrast, the diffraction patterns of the FCC fines reference material and amoxicillin-loaded FCC fines display only 15–20 significant peaks in this region, suggesting a lower degree of crystallinity.

In fact, most of the features above 15 nm⁻¹ are associated with the FCC fines reference material, while the features below 15 nm⁻¹, about 5–6 minor peaks, are associated with the loaded amoxicillin. This indicates that the amoxicillin loaded onto the particles has very limited long range crystalline order [36]. Generally, amorphous materials have no peaks in diffraction patterns. However, it must be remembered that the material mass and electron density of amoxicillin in comparison to FCC is on average considerably less. The intensity of the visible features may, nonetheless, help elucidate whether the loaded amoxicillin is amorphous or manifesting an alternative structure, such a nanocrystals or a liquid crystal mesophase. Both amorphous and nanocrystalline phases contain no long-range order, meaning that there are no coherent multiple regular crystalline planes to diffract X-rays. For amorphous materials the incident X-rays are scattered

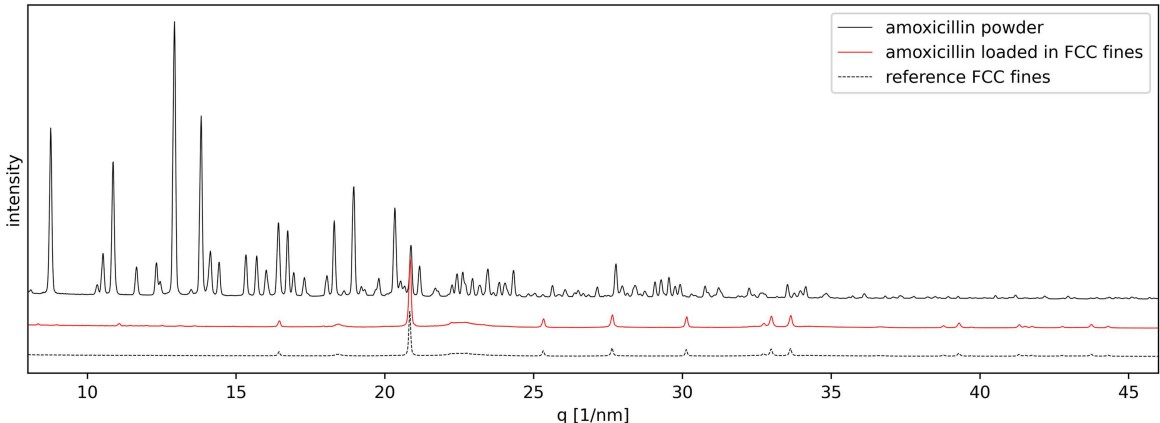

**Fig 5. Diffraction pattern for amoxicillin loaded in FCC fines (continuous red curve), amoxicillin powder (continuous black curve) and FCC fines reference (dashed black curve).** Curves have been offset by a constant value for easier visualization.

isotropically and there are no sharp peaks in the diffraction pattern. In other words, long range ordered crystalline parts give sharp narrow diffraction peaks and a truly amorphous component would give a broad background contribution [36], which is usually termed a "halo". The lack of sharp crystalline peaks or a significant amorphous halo cannot be attributed to an exceedingly weak interaction of X-rays with the samples. Thus, it might be more appropriate to consider the lack of an amorphous background in the spectrum in this case as an indication rather of nanocrystallites (lacking longer range order, with exceptionally high surface area) or a mesophase state exhibiting order in only one (nematic) or maximally two dimensions (smectic). These more complex structures could account for both the lack of diffraction peaks and the absence of amorphous background together with a high rate of solubility related to either the extremely high nanocrystalline or mesophase surface area.

The significant loss of crystallinity is evidenced by the 25-fold reduction in peak intensity when amoxicillin is loaded into the pores and onto the surfaces of FCC fines. Given that the total amount of the amoxicillin exposed in the two experiments was the same (pure amoxicillin and amoxicillin loaded into the FCC fines) and the fact that the signal intensity is proportional to the amount of crystalline material, a 30 w/w% decrease in material should result in only a one-third reduction in signal intensity, not a 25-fold decrease [37,38]. This stark reduction indicates a substantial loss of long-range crystallinity. Similar reductions in crystallinity were observed in other samples, including FCC fines loaded with 15 w/w% and 30 w/w% amoxicillin and FCC granules loaded with 15 w/w% amoxicillin, respectively (data available in the Supporting Information).

Given that the XRD diffractogram of the loaded amoxicillin did not display the characteristic amorphous "halo" but a significant reduction in peak intensity compared to pure amoxicillin, the structural state of the amoxicillin within the porous material could not be conclusively determined. To investigate further whether the loaded amoxicillin in the confined space of the porous FCC material is amorphous, nanocrystalline or mesophase, we conducted experiments with higher sensitivity and spatial resolution using the hard X-ray nanoprobe beamline, NanoMAX, at MAX IV in Lund, Sweden.

## Synchrotron-based XRD experiment

For the synchrotron-based XRD experiments, four different samples were analyzed: (i) amoxicillin powder, (ii) amoxicillin loaded into FCC fines (30 w/w%), (iii) unloaded FCC fines, and (iv) reference amoxicillin precipitated by solvent evaporation. As expected, the diffractogram of the pure amoxicillin powder (continuous black line in Fig 6A) exhibits intense peaks, characteristic of its crystalline structure. Conversely, the amoxicillin precipitated by solvent evaporation (dashed red line

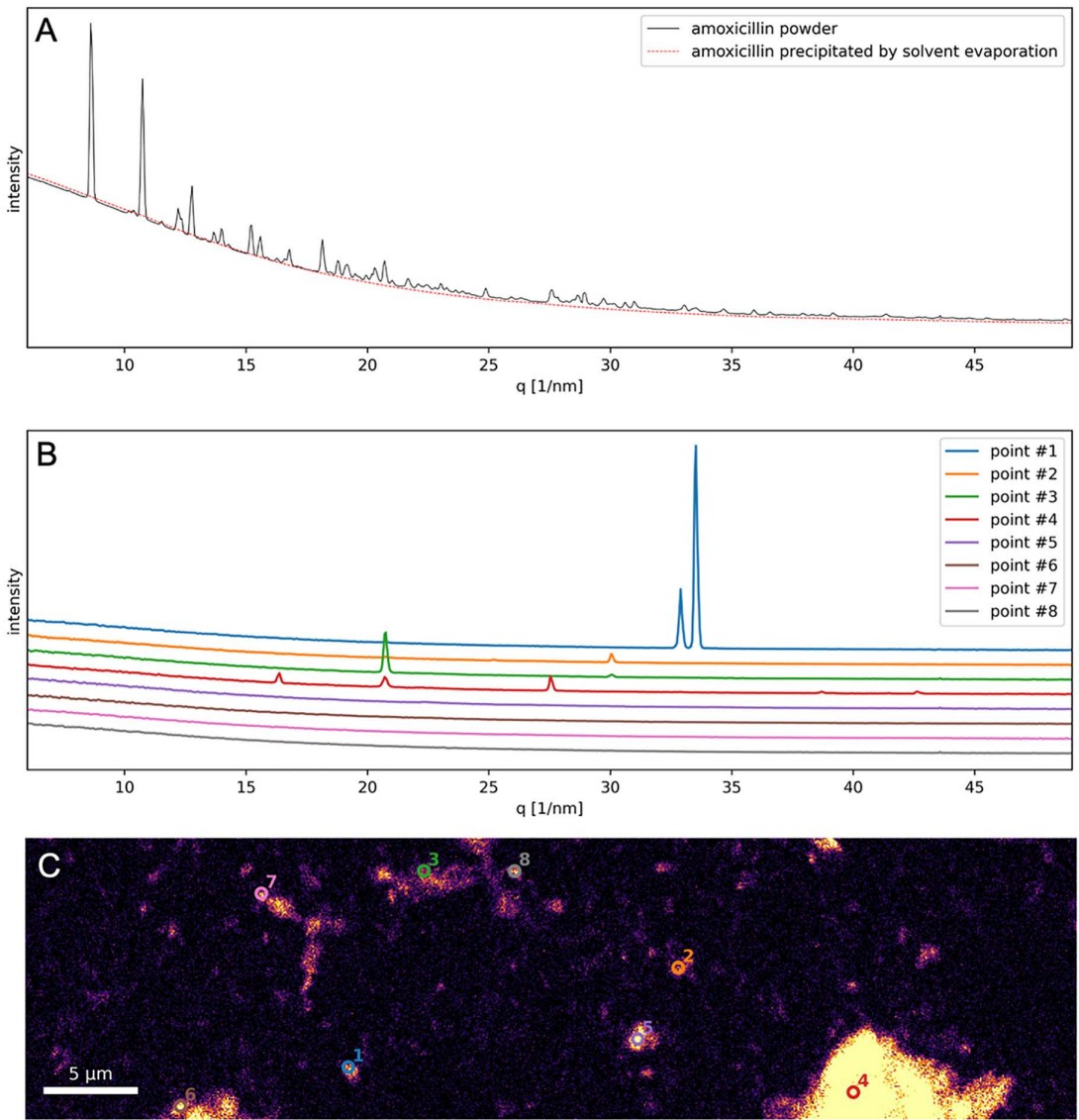

**Fig 6. Normalized diffractograms of pure amoxicillin as compared to amoxicillin precipitated by solvent evaporation (A) and loaded in FCC fines (B); diffractograms in (B) have been offset by a constant value for easier visualization.** High resolution map of the amoxicillin loaded in FCC fines **(C)**: bright pixels indicate the presence of amoxicillin. The diffractograms displayed in (B) are extracted from the points annotated in **(C)**.

in Fig 6A displays no discernible peaks, confirming a long-range disordered state. To facilitate comparison, each diffractogram was rescaled by dividing it by its average intensity within the low $q$ range (9–13 nm$^{-1}$) preceding the region featuring diffraction peaks. This normalization enables a clearer comparison of the structural features between the samples.

The diffractograms from amoxicillin loaded in FCC fines exhibited significantly lower intensity features and only for a few scanning positions (see points 1–4 in Fig 6B-C). This suggests a predominant loss of crystallinity in the amoxicillin loaded into the particles. The bright regions from Fig 6C correspond to S-rich regions and therefore indicate the presence of amoxicillin: an inhomogeneous distribution with clusters of variable size is apparent. Interestingly, most pixels revealing a significant presence of amoxicillin – of which only 8 selected examples are displayed in Fig 6B – correspond to

diffractograms without visible diffraction peaks thus indicating that the majority of the amoxicillin present in this sample is in an amorphous state.

## Scanning Electron Microscopy

The SEM images presented in Fig 7 provide further support for the findings from the amoxicillin release studies and the X-ray experiments. The pure amoxicillin exhibits elongated, rod-like crystalline formations, which resemble elongated prismatic shapes. In contrast, the amoxicillin precipitated within the pores or on the surface of the FCC fines displays a distinct snowflake-like structure.

The SEM images in Fig 7 illustrate distinct precipitation morphologies: pure amoxicillin shows elongated crystalline rods (Fig 7C), whereas amoxicillin loaded into FCC fines (Fig 7B) forms irregular, sheet-like domains both within and on the surface of the porous matrix. This suggests that confinement within the particle pores restricts long-range crystal growth, consistent with the pseudo-amorphous state observed by XRD. The coexistence of intra-pore amorphous deposits and surface nanocrystals likely governs the biphasic release behavior: rapid dissolution of surface-deposited amoxicillin accounts for the initial burst, while the confined, less soluble amorphous fraction contributes to the sustained release over months. Because amorphous material generally exhibits higher apparent solubility and faster dissolution than crystalline drug, the intrapore amorphous fraction is expected to dissolve readily once water penetrates the FCC pores, whereas the pre-existing surface nanocrystals dissolve more slowly. This structural heterogeneity therefore provides a straightforward explanation for the biphasic release observed.

This morphological difference suggests that the amoxicillin, when loaded into the FCC fines during the solvent evaporation process, precipitates in an amorphous or 2D nanocrystallite-composite sheet state. This altered structural state is likely a key factor contributing to the observed enhanced dissolution rate of the drug. We can speculate that the driving mechanism for nanocrystal formation could be related to a sorption effect of molecules in contact with a surface which could provide a nanoscale molecular orientation effect, a conformation that prevents the lowest energy intermolecular structure, namely preventing crystal formation. Because the FCC particles contain a high fraction of nanoscale pores, the amoxicillin can only adopt a locally stable, size-restricted crystal form; nanoscale confinement therefore limits crystal growth and stabilizes nanocrystals even though the bulk crystalline state would represent the global thermodynamic minimum.

## Release of amoxicillin from the EC films in water

From the wide range of polymers suitable for a polymer matrix, we selected ethyl cellulose (EC), a non-biodegradable, biocompatible polymer that is one of the most studied encapsulating materials for controlled drug release [13,39–42]. Although cellulose and its derivatives are environmentally friendly and actively degradable by various bacteria and fungi, Marston *et al*. demonstrated that cellulose is a biodurable material when implanted in animal and human tissues, as resorption does not occur due to the absence of cellulase synthesis in cells [43–45]. Please note that for these experiments we used the Omyapharm particles – which are the starting material for the FCC fines and granules.

Depending on the intended application, the amoxicillin release can be adjusted in various ways: by tuning the amoxicillin loading level in the porous particles, changing the amount of particles in the EC solution formulation, as well as by inclusion of additives. In this proof of concept experiments we show the release of amoxicillin at a specific surface to water exposure. The release studies from the Omyapharm–EC films were performed under non-sink conditions, intentionally chosen to better represent diffusion-limited environments relevant to indwelling medical devices, such as catheter surfaces. In such applications, fluid exchange is restricted and local concentration gradients develop over time, making the non-sink setup a more realistic analogue of the in-use situation.

In this case a rolled EC film was immersed in water where the surface to water exposure was set to 5.4 cm$^2$/mL, as described in the Materials section. To evaluate the advantage of loading amoxicillin in porous particles and the influence of an additional loading step, the EC film was compared to a reference EC film containing a physical mixture of amoxicillin

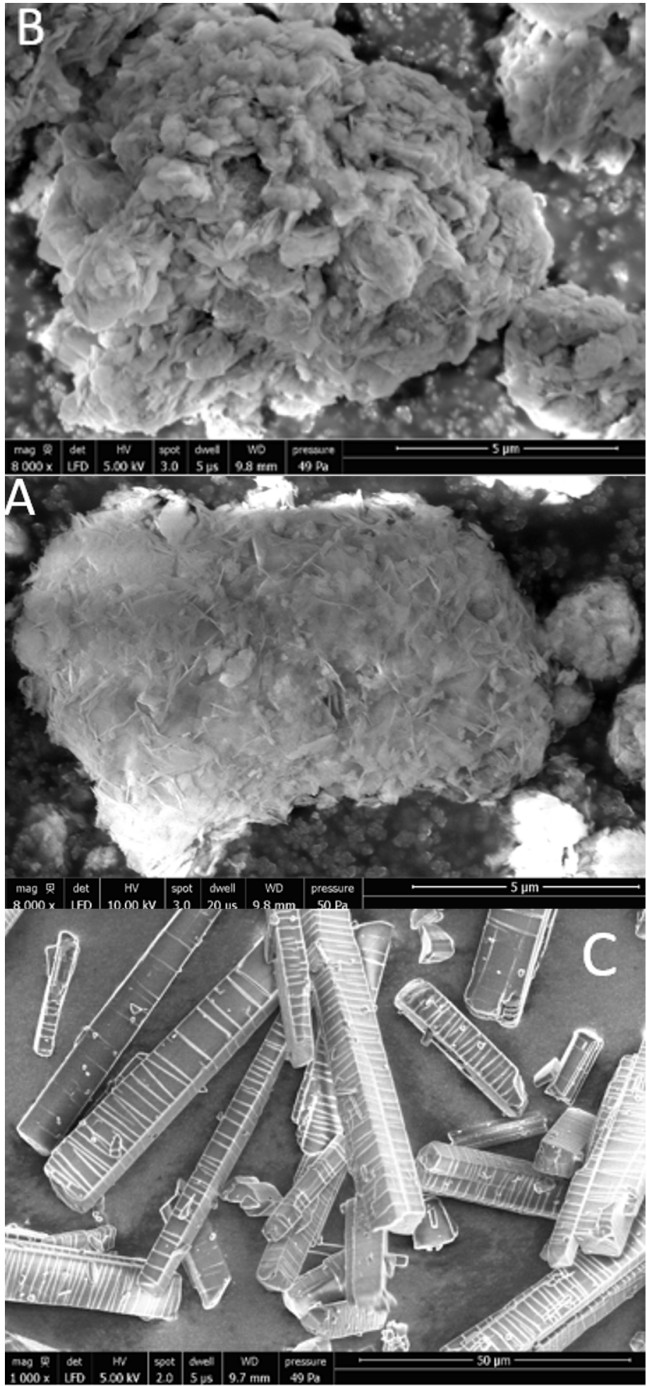

**Fig 7. SEM images of a) FCC fines, b) 15 w/w% amoxicillin-loaded FCC fines, and c) powder amoxicillin.**

and Omyapharm particles where the amoxicillin and Omyapharm were added separately into the polymer solution, as seen in Fig 8. As mentioned in Materials section Omyapharm particles are known to have the same structural and chemical properties as FCC fines and granules. Note that the ratio of all components was kept identical.

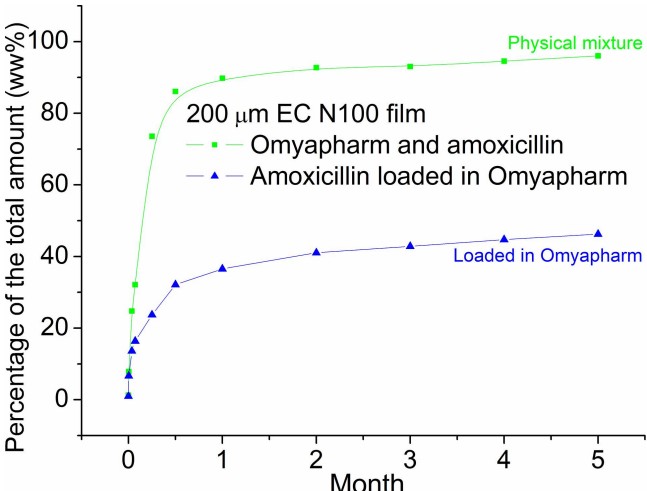

**Fig 8. Amoxicillin release over time from the following films: Omyapharm particles loaded with amoxicillin (blue), Omyapharm particles and amoxicillin mixture (green) – as a reference.** Data shown represents a single representative experiment for each film.

In the release experiments, the amoxicillin concentrations were measured after 2 h, 1 day and 2 days, 1 week, 2 weeks 4 weeks and then monthly up to 5 months. As shown in Fig 9, both release profiles show burst behavior over the first approximately 15 days reaching 85% for unloaded amoxicillin (physical mixture) and 39% for loaded amoxicillin, respectively. After approximately 1 month, both profiles show similar behavior with a slight monotonous increase in kinetics. At the end of the experiment (5 months), unloaded amoxicillin was almost completely released (96%) whereas films with loaded amoxicillin had only released 46% of the amoxicillin. Please note that this long-term release studies reported here were performed on single representative samples and statistical replication was not feasible for this multi-month proof of concept experiment. Future work will include replicated experiments to provide statistical validation and kinetic model fitting.

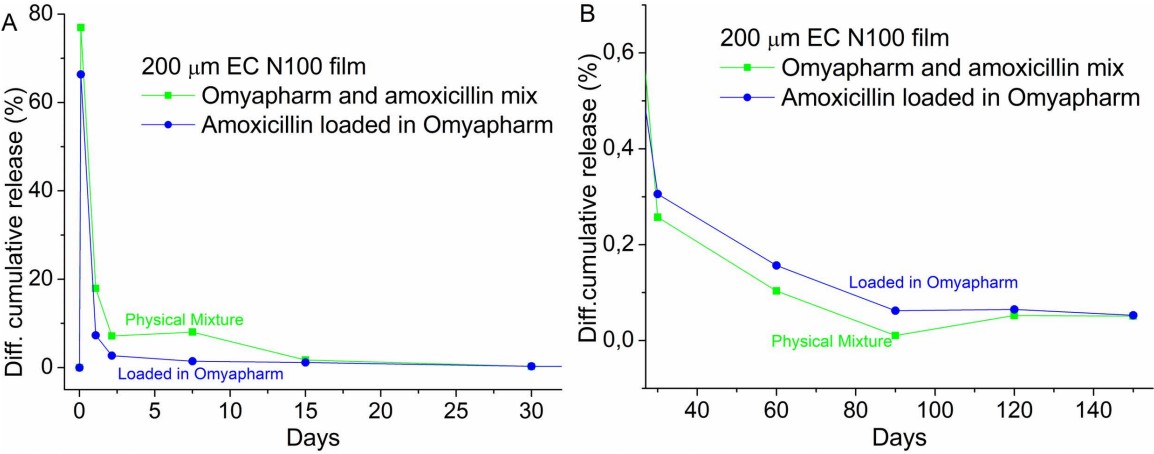

**Fig 9. The differential release profile of the films made from amoxicillin loaded in Omyapharm particles (blue) and Omyapharm particles and amoxicillin mixture (green) for A) month 1 and B) month 2-5.**

By closely looking at the release rate, it can be noted that in the second phase of the release (month 2–5) amoxicillin loaded into Omyapharm is released at a higher rate than the unloaded part. During the initial burst phase (month 1), the estimated release rate is 13%/month for the loaded amoxicillin and 19%/month for the unloaded amoxicillin, respectively. During the second phase (month 2–5), the estimated release rate is 0.032%/month for the loaded amoxicillin and 0.023%/month for the unloaded amoxicillin, respectively. In other words, the sustained release appears to be a result of a reduced burst rather than a slower diffusion of amoxicillin through the EC film. Loading actives into porous Omyapharm particles prior to the integration into a polymer matrix might therefore be a viable approach to mitigate extensive burst of actives from thin films. Although we cannot exclude minor recrystallization during the multi-month release experiment, the observed long-term profile is most consistently explained by the pre-existing coexistence of surface nanocrystals and intrapore amorphous amoxicillin. The confined pore geometry strongly limits any significant crystal growth during dissolution, making crystallization during release unlikely to be a major contributor.

Albeit a different dosage form, this time frame is also much longer compared to the Singh *et al.* study [5], where release of amoxicillin from microspheres was sustained by/through a mucoadhesive device. Their best formulation, based on a combination of HPMC K4M as mucoadhesive and Eudragit RS 100 as polymer matrix, showed a continuous release over a maximum of 12 h only in-vitro, with the release profile reaching 96.15 w/w% of the total amount in that relatively short period. Additionally, Vafayi *et al.* [10] showed that porous hollow silica nanospheres ZnS@SiO2 loaded with amoxicillin showed a delayed release pattern in simulated body fluid with 84% of the total amoxicillin released over a 24 h period. Also, compared to the MCM-41 [8] and Mg-HA carriers systems [9], which in best cases achieved up to 13% amoxicillin release within 24 hours, and 89% within 40 hours, respectively; our FCC–ethyl cellulose composite films demonstrate a markedly prolonged release.

Similarly, Torres-Figueroa *et al.* [6] have studied the effectiveness of composite hydrogels – poly(acrylamide) and starch – as a platform for controlled release of amoxicillin, however the hydrogel reported released 48.6 w/w% of the amoxicillin load within the first 24 h. This behavior was attributed to the hydrophilic character of the starch and the swelling rate of the hydrogel affecting the kinetics of the drug release. Also, Bohns *et al.* investigated the incorporation of amoxicillin-loaded polymeric microspheres into mineral trioxide aggregate (MTA) cement formulations for dental applications. Their best formulation achieved a release of 16% amoxicillin within 96 h. The longest sustained drug release we found in the literature was obtained by Prasanna *et al.* [11], who studied the release of amoxicillin from a hydroxyapatite layer-by-layer coated with polyvinyl alcohol and sodium alginate. Apart from a good antibacterial activity they showed a sustained release of amoxicillin for about 30 days. In none of these studies did the loading/impregnation of amoxicillin change the physical state from the original crystalline form, in contrast to where we see the loading of Omyapharm particles resulting in a change from long range crystalline to a pseudo-amorphous state.

The amoxicillin release depends on the hydrophilic character of the used polymer. HPMC [5], poly(acrylamide), starch [6] or polyvinyl alcohol [11] have been reported examples. In our case, although EC is not water soluble, it is nonetheless hydrophilic. In spite of the insolubility of EC, the amoxicillin finds its way out to the surface along hydrophilic fissures, most likely formed during the initial step of solvent evaporation that takes place during film formation. Relying on such fissures would not be a guarantee for successful delivery.

Another more likely possibility is that water molecules, under osmotic pressure, can diffuse into the molecular-sized spaces of EC polymer and reach amoxicillin loaded particles. Thus, the amoxicillin solution formed within the film can diffuse out under the resulting concentration gradient. Even though the amoxicillin loaded in the porous particles have a higher water solubility (dissolution rate) as compared to pure amoxicillin, the results of the tempered release rate of the loaded amoxicillin can be mainly explained by the significantly reduced initial burst. However, additional contribution may come from other factors such as: i) additional tortuosity provided by the particle; ii) sedimentation of the loaded particles (higher density provided by the porous particles) during film production and hence longer diffusion pathway through the EC matrix; iii) easier access to the more homogeneously distributed and smaller pure amoxicillin particles as compared to

more concentrated amoxicillin – porous particles loaded domains, which, in turn, can be more effectively encapsulated by the EC.

To sum up, compared to previously reported amoxicillin carriers such as MCM-41 mesoporous silica (13% release within 24 h), Eudragit S100 microparticles (16% within 96 h), and Mg–HA porous ceramics (89% within 40 h), the FCC–EC composite films developed in this study exhibit a uniquely sustained release profile spanning several months. The combination of FCC's hierarchical porosity with the diffusion-modulating ethyl cellulose matrix provides a synergistic route to long-term antibiotic delivery, not previously reported for amoxicillin formulations.

The results obtained in this study suggest that a more efficient system in terms of controlled delivery can be obtained by loading amoxicillin into porous microparticles – functionalized calcium carbonate – instead of mixing it directly into a polymer matrix; and then homogeneously distributing these particles within a polymer matrix. Reducing initial burst avoids unnecessarily high concentrations and allows for more judicious release rates over a prolonged period of time. It should be noted that this work represents a proof-of-concept demonstration focused on the sustained release performance of FCC–EC films rather than on their mechanical or interfacial characterization. For future development toward indwelling medical applications, complementary evaluation of tensile strength, swelling behavior, and mucoadhesive properties are essential and it will be considered in the future.

The observed release concentration of amoxicillin (above 0.1 mg/mL) exceeds the minimum inhibitory concentration (MIC) for E. coli, which is 0.008 mg/mL [46,47]. Eventually, the release rates of amoxicillin from ethyl cellulose films can be tailored by manipulating the preparation method, film thickness, and the inclusion of specific additives in order to achieve relevant performance as drug delivery systems in catheter and indwelling applications. This comparison is intended as contextual guidance only and does not imply predictive relevance for in vivo antimicrobial efficacy.

## Conclusions

Since bacterial resistance to antibiotics is recognized currently as a major threat to global health, it is desirable to change the way antibiotics are used, crucially supporting their judicious application. Today, already established infections can only be treated with selected antibiotics, and it is important to use as little as possible to minimize antimicrobial resistance, but the key point is that this controlled small amount must be maintained for long periods above the minimum inhibition concentration (MIC). Therefore, sustained release systems are regarded as a viable solution to maintaining microbial-free treatment and, hence, reducing the chance of antimicrobial resistance.

Within this work, a model drug – amoxicillin, was successfully loaded into porous functionalized calcium carbonate particles using a solvent evaporation method achieving drug loads of 30 w/w%. Dissolution of loaded amoxicillin was faster compared to pure amoxicillin as demonstrated by UV-vis spectrometry. This can be explained by the inhibition of crystal formation during the loading process as supported by XRD and SEM measurements. Structural analysis reveals the presence of a pseudo-amorphous, nanocrystalline or molecularly surface oriented mesophase layer form. However, a total area of only 2880 $\mu m^2$ was investigated by nanoscale XRD within this study, therefore the reported observations are only qualitative interpretations. Further measurements of larger areas and on more samples could produce more statistics, thus leading to quantitative results, such as the relative abundance of nanocrystalline or mesophase areas with respect to amorphous ones as well as the average size of nanocrystals or mesophase layers.

If this essentially non-crystalline material is embedded in a biocompatible polymer, such as ethyl cellulose (EC) – a flexible water insoluble film that can be easily manipulated, could be produced. With this film construct we have shown that the amoxicillin release can be sustained for more than 5 months. The release through the EC matrix of amoxicillin from loaded particles may effectively mitigate the issue of burst release by minimizing the initial surge and ensures a sustained release of the active ingredient over a prolonged period, reducing wastage during the early phase. In other words, the same amount of amoxicillin can be delivered over a much longer period of time, rendering this film well suited, for example, in catheter/indwelling applications.

## Supporting information

**S1 Fig. Calibration curve of the released amoxicillin in water obtained by using the UV-vis absorbance at 272 nm.**
(TIF)

**S2 Fig. The amoxicillin release curves in water from 30% amoxicillin loaded FCC granules.**
(TIF)

**S1 Table. Residual Amoxicillin loaded in FCC fines measured with TGA (up to 600*C) after release in water.**
(DOCX)

**S3 Fig. UV–Vis absorbance spectra of FCC fines, FCC granules, and Omyapharm in water, showing no absorbance near 272 nm (Spectra were corrected for pure water absorbance).**
(TIF)

**S4 Fig. Amoxicillin release profiles for Omyapharm-loaded ethyl cellulose (EC) films of different thicknesses (200 μm and 160 μm).** The thinner film (160 μm) exhibits a comparable biphasic release pattern—an initial burst followed by a slower sustained phase—demonstrating reproducibility of the release behavior.
(TIF)

**S2 Table. Summary of experimental conditions for particle and film release studies.**
(DOCX)

**S1 Text. Applicability of classical kinetic models.**
(DOCX)

## Acknowledgments

Ulf Johansson is gratefully acknowledged for support during the XRD experiment at NanoMAX.

## Author contributions

**Conceptualization:** Petru Niga, Simone Sala, Jenny Rissler, Lina Nyström, Anna Fureby, Ulla Elofsson, Roger Roth, Joachim Schoelkopf, Patrick Gane.

**Data curation:** Petru Niga, Simone Sala, Lina Nyström, Ulla Elofsson, Roger Roth.

**Formal analysis:** Simone Sala, Jenny Rissler, Lina Nyström, Anna Fureby, Ulla Elofsson, Roger Roth.

**Funding acquisition:** Petru Niga, Anna Fureby, Ulla Elofsson.

**Investigation:** Lina Nyström, Joachim Schoelkopf, Patrick Gane.

**Methodology:** Simone Sala, Roger Roth, Patrick Gane.

**Project administration:** Petru Niga.

**Supervision:** Petru Niga.

**Visualization:** Petru Niga, Simone Sala, Jenny Rissler.

**Writing – original draft:** Petru Niga, Simone Sala, Jenny Rissler, Lina Nyström, Anna Fureby, Ulla Elofsson, Roger Roth, Joachim Schoelkopf, Patrick Gane.

**Writing – review & editing:** Petru Niga, Simone Sala, Jenny Rissler, Lina Nyström, Ulla Elofsson, Roger Roth, Joachim Schoelkopf, Patrick Gane.

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
