## [Decision Letter · Decision Letter 0]

7 Apr 2025

Dear Dr. Niga,

Thank you for submitting your manuscript to PLOS ONE. After careful consideration, we feel that it has merit but does not fully meet PLOS ONE’s publication criteria as it currently stands. Therefore, we invite you to submit a revised version of the manuscript that addresses the points raised during the review process.

**One or more of the reviewers has recommended that you cite specific previously published works. Members of the editorial team have determined that the works referenced are not directly related to the submitted manuscript. As such, please note that it is not necessary or expected to cite the works requested by the reviewer**

We look forward to receiving your revised manuscript.

Kind regards,

Mahmood Ahmed

Academic Editor

PLOS ONE

[This work was supported by Omya International AG and RISE Research Institute of Sweden. We acknowledge MAX IV Laboratory for time on Beamline NanoMAX under Proposal 20230043. Research conducted at MAX IV, a Swedish national user facility, is supported by the Swedish Research council under contract 2018-07152, the Swedish Governmental Agency for Innovation Systems under contract 2018-04969, and Formas under contract 2019-02496.].

Additional Editor Comments (if provided):

Reviewers' comments:

Reviewer's Responses to Questions

**Comments to the Author**

1. Is the manuscript technically sound, and do the data support the conclusions?

Reviewer #1: Partly

Reviewer #2: Partly

Reviewer #3: Yes

2. Has the statistical analysis been performed appropriately and rigorously?

Reviewer #1: No

Reviewer #2: No

Reviewer #3: Yes

3. Have the authors made all data underlying the findings in their manuscript fully available?

Reviewer #1: No

Reviewer #2: Yes

Reviewer #3: Yes

4. Is the manuscript presented in an intelligible fashion and written in standard English?

Reviewer #1: Yes

Reviewer #2: Yes

Reviewer #3: Yes

Reviewer #1: Dear Authors,

The research work is quite interesting with well problem statement but authors have designed the experiment for verifying material properties of amoxicillin loaded FCC not in a good manner and not thorough study to clearly understand the characteristic of the loaded FCCs. It is very difficult to follow mentioned experimental results step by step. It is very regret to recommend for rejection due to the quality of presented experimental data especially in the part of material characterization. For example:

1. To study the releasement of amoxicillin from FCC, FCC fines and FCC granules were used as carrier but to study the releasement of the amoxicillin from the Ethyl cellulose film, Omyapharm FCC500-OG was used instead without notifying the solid support to convince the possibility. Only particle size ranges of FCC fines and FCC granules were provided but not BET specific area not bulk density.

2. The content of amoxicillin is not informed for releasement study from the loaded FCC granules (Figure 4) but from the loaded FCC fines, the releasement results is presented both from 15% and 30% amoxicillin loading. The releasement of neat FCC is not comparatively studied.

3. XRD diffraction pattern of precipitated amoxicillin is presented but its SEM micrograph is not shown, to verify different morphology.

4. In the material characterization results from loaded FCC, the finding are rarely compared to other similar finding and made discussion.

Reviewer #2: The study presents an interesting approach utilizing functionalized calcium carbonate (FCC) microparticles embedded in ethyl cellulose (EC) films for sustained antimicrobial drug release. This research is particularly relevant for indwelling medical applications, addressing the need for prolonged drug release in such devices.

1. Please explain the novelty compared to published work in this field. Pure drug used or its salt used?

2. The introduction can be improve with recent work in this filed, like, https://doi.org/10.1007/s12668-024-01722-3, https://doi.org/10.2174/0115748855274979231228103038, https://doi.org/10.1007/978-981-99-1655-9_6, https://doi.org/10.1016/j.foodhyd.2020.106206, https://doi.org/10.1016/j.jdent.2024.105489, https://journals.lww.com/jpbs/toc/2012/04001 etc

4. No clarity on how formulation optimized?

5. How , XRD, To understand the underlying mechanisms influencing amoxicillin release, give this understanding?

6. How author is sure that during casting in organic solvent, microparticle not dissolve/break?

7. Why HPMC K4m and HPMC K4M and Eudragit RS 100 selected as film former?

8. The characterization of film like, Muco adhesion, tensile strength, swelling etc should be compiled.

Reviewer #3: Comment 1: Keep the term fixed either extended or sustained release system.

Comment 2: Can you check this line ‘an increased release rate from the particles compared to pure amoxicillin powder’ release rate will be extended? And also make it which on e is considered here a paprticle?

Comment 3: sustained 3drug delivery acts to release the drug at a controlled rate’ There is a difference between sustained and controlled release

Comment 4: significant efforts are directed at developing new indwelling urinary catheters (IUCs)’ discuss related works on this. See such articles Int J Biol Macromol, 162, 2020, 1849-1861. Polymers. 2022; 14 (4): 746; J. Drug. Deliv. Sci. Tech., 72, 2022, 103380

Comment 5: mention the molecular weight of EC and the subsection process is not proper.

Comment 6: the amoxicillin must first be dissolved in a solvent with a low boiling point. ‘ Mention the solvent

Comment 7: amoxicillin release in water was determined using a thermogravimetric analyzer’ This procedure is not proper. Please cite some related works if anyone has followed the method.

Comment 8: If XRD procedure is needed in so details

Comment 9: Calibration curve can be added

Comment 10: Please check Figure 5

.

Reviewer #1: No

Reviewer #2: No

Reviewer #3: No

---

## [Author Response · Author response to Decision Letter 1]

3 Jun 2025

Reviewer #1:

Dear Authors,

The research work is quite interesting with well problem statement but authors have designed the experiment for verifying material properties of amoxicillin loaded FCC not in a good manner and not thorough study to clearly understand the characteristic of the loaded FCCs. It is very difficult to follow mentioned experimental results step by step. It is very regret to recommend for rejection due to the quality of presented experimental data especially in the part of material characterization. For example:

The authors thank the Reviewer for the opinion regarding the worth of the work and acknowledge the aspects of concern. We hereby undertake to respond to the comments as follows:

1. To study the releasement of amoxicillin from FCC, FCC fines and FCC granules were used as carrier but to study the releasement of the amoxicillin from the Ethyl cellulose film, Omyapharm FCC500-OG was used instead without notifying the solid support to convince the possibility. Only particle size ranges of FCC fines and FCC granules were provided but not BET specific area not bulk density.

To clarify: the material family of functionalised calcium carbonate (FCC) retains consistent structural and chemical properties irrespective of sample name definition. We now have put more emphasis to this in the new manuscript text.

The use of a fractionated sample of FCC, in respect to fines and granules, was designed to study the impact of individual dispersed particle size versus structuration of those individual particles to form granules. The granular material has been shown earlier to accelerate advantageously pharmaceutical tablet disintegration in aqueous medium, for example, as illustrated in these extra references, which have now been added to the manuscript:

D. Preisig, et al., ‘Drug Loading into Porous Calcium Carbonate Microparticles by Solvent Evaporation,’ Eur. J. Pharm. Biopharm. 87, 548–558 (2014).

T. Stirnimann, et al., ‘Compaction of Functionalized Calcium Carbonate: A Porous and Crystalline Microparticulate Material with a Lamellar Surface,’ Int. J. Pharm. 466(1–2), 266–275 (2014).

T. Stirnimann, et al., ‘Functionalized Calcium Carbonate as a Novel Pharmaceutical Excipient for the Preparation of Orally Dispersible Tablets,’ Pharm. Res. 30(7), 1915–1925 (2013).

The context of exemplifying structuration of FCC in respect to release kinetic of amoxicillin is, therefore, undertaken here to establish if such structuration provides any release mechanistic advantages, stressing once again throughout that the particle structure and chemical nature of the particles remain intrinsically constant.

Omyapharm FCC500-OG illustrates the current commercially available product from which the experimental fractions were similarly derived. This text is now included in the manuscript.

Nitrogen adsorption analysis in terms of BET specific surface measurement and bulk density are now also reported.

2. The content of amoxicillin is not informed for releasement study from the loaded FCC granules (Figure 4) but from the loaded FCC fines, the releasement results is presented both from 15% and 30% amoxicillin loading. The releasement of neat FCC is not comparatively studied.

The loading fraction of amoxicillin was consistently 30 w/w% and 15 w/w% for all samples. Section 2.2. was unfortunately somewhat misleading in this respect as the term FCC particles was specifically used to refer generically to all sample forms. The text is now modified to confirm the loading across all FCC sample types. We understand the implication raised by the reviewer in that granulated material has an additional pore space, i.e. between the individual particles, into which the amoxicillin will also enter. This is now also expressed in the manuscript as follows:

[2.2. Particle Loading

To load amoxicillin into the porous FCC particles, the amoxicillin must first be dissolved in a

solvent with a low boiling point and associated high vapour pressure. This solution is then mixed with the FCC particles as present in all structural forms used in the study, namely FCC fines, granules and commercial Omyapharm, and the solvent is allowed to evaporate. During evaporation, amoxicillin precipitates both within the pores and on the surface of the FCC particles. In this study, a 3 g/L solution of amoxicillin in an ethanol/acetone mixture was prepared. This solution was then added to a predefined amount of particles (for 30 w/w% and 15 w/w % loading – 7 g FCC and 8.5 g FCC, respectively, per 1 L solution) and the solvent was evaporated using a rotary evaporator (30 °C and 400-150 mbar). It is recognised that the distribution of amoxicillin in the case of the granulated structure will include loading between particles, and not only on the surface, and within the pores of the individual particles. Given the nature of the granule concept, material between the particles, to a first approximation, is likely to behave similarly to that associated with the exterior of the particles, though it should be recognised that access of liquid to the interior of a granule may be hindered to reduced permeability. Such effects reveal themselves during the release experiment study.]

We appreciate the reviewer’s suggestion regarding the inclusion of a neat FCC control. However, as the release studies were based on UV-Vis spectroscopy at 272 nm—a wavelength specific to amoxicillin—neat FCC does not exhibit any absorbance at this wavelength. Therefore, it does not contribute to the measured signal, and its inclusion as a control would not affect the release data. This point is now clarified in the revised manuscript.

3. XRD diffraction pattern of precipitated amoxicillin is presented but its SEM micrograph is not shown, to verify different morphology.

The morphology of amoxicillin depends on two main factors, the dynamic of precipitation and the surface onto which the precipitate forms. Amoxicillin in free solution under slow evaporation leads to microscopic single crystals, however under induced rapid precipitation, such as by the presence of a surface or nucleation centres, naturally tends to be amorphous. When considering monomolecular layers, such as those deposited on the surface of the FCC particle pores, microscopic imaging, by definition, unfortunately fails to reveal such molecular structure information. It is for this reason that resort is made to XRD diffraction, as reported in the manuscript, it being sensitive to molecular and intermolecular structure of relevance here.

4. In the material characterization results from loaded FCC, the finding are rarely compared to other similar finding and made discussion.

We appreciate the Reviewer’s suggestion to broaden the contextual discussion of amoxicillin release from particulate systems beyond FCC, and we have now included comparative insights from other porous carrier systems. For example, Li et al. (2010) reported on organically modified MCM-41 mesoporous silica materials for amoxicillin delivery, where drug release was about 13% within 24 hours. Also, we reference studies such as Vafayi et al., who demonstrated a three-stage release profile from hollow silica nanospheres in simulated body fluid, and Bohns et al., who evaluated amoxicillin-loaded Eudragit S100 microparticles in dental applications. Their best formulation achieved a release of 16% amoxicillin within 96 hours. Additionally, Wang et al. who worked on the release of amoxicillin from Mg-HA porous particles. In this work 89% of the amoxicillin was released within 40hours. These examples have been incorporated into the revised manuscript to contextualize our results within the broader field of porous drug delivery systems. In contrast, our FCC-ethyl cellulose film composites demonstrate markedly extended release, supporting applications requiring long-term antimicrobial activity.

Li et al., J. Colloid Interface Sci., 342 (2010), 607–613, https://doi.org/10.1016/j.jcis.2009.10.073.

L. Vafayi et al., Bioinorganic Chemistry and Applications 2013 Vol. 2013 Issue 1 Pages 541030 https://doi.org/10.1155/2013/541030

Bohns et al., Restor Dent Endod 2020 Vol. 45 Issue 4 Pages 50. DOI: 10.5395/rde.2020.45.e50

Wang et al., Materials and Manufacturing Processes 2022 Vol. 37 Issue 13 Pages 1500-1505 DOI: 10.1080/10426914.2021.2016824

Reviewer #2:

The study presents an interesting approach utilizing functionalized calcium carbonate (FCC) microparticles embedded in ethyl cellulose (EC) films for sustained antimicrobial drug release. This research is particularly relevant for indwelling medical applications, addressing the need for prolonged drug release in such devices.

The authors are grateful to the reviewer for recognising the value of the main target application for this work.

1. Please explain the novelty compared to published work in this field. Pure drug used or its salt used?

In many cases, pure drugs tend to express hydrophobicity due to their organic molecular constituents. The advantage of depositing a monomolecular layer is that the dynamic of this arrangement is constrained prior to exposure to water, and the rapid dissolution thereby prevents the macro-induced hydrophobicity. The citation regarding the delivery of nutraceuticals and flavourings (11) (Lundin Johnson et al.) illustrates this nicely. In the case of amoxicillin sodium salt dissolution is not normally impaired by any hydrophobic effects, and, as the Reviewer correctly suggests, loading on porous media is not a prerequisite to ensure dissolution. The target here is to retain a reservoir of the amoxicillin whilst at the same time supporting a continuous release of the active substance at a level above the minimum bacterial inhibition concentration to ensure a continuous antibacterial activity in applications prone to recurring infection.

2. The introduction can be improved with recent work in this filed, like, https://doi.org/10.1007/s12668-024-01722-3, https://doi.org/10.2174/0115748855274979231228103038,

https://doi.org/10.1007/978-981-99-1655-9_6,

https://doi.org/10.1016/j.foodhyd.2020.106206,

https://doi.org/10.1016/j.jdent.2024.105489,

https://journals.lww.com/jpbs/toc/2012/04001 etc

We thank the Reviewer for the suggested literature. We have carefully reviewed the references provided and acknowledge their contribution to the broader field of drug delivery. However, these works focus on different types of materials or delivery systems that do not closely align with the specific objective of our study—namely, the integration of drug-loaded FCC particles into ethyl cellulose films for sustained release. To maintain a focused and concise introduction that accurately reflects the scope of our work, we have decided not to include these particular references. Nevertheless, we appreciate the Reviewer’s effort in highlighting them.

4. No clarity on how formulation optimized?

The formulation per se has two interactive aspects, (i) the dissolution and rapid evaporation properties required when loading the porous medium particles, and (ii) the distribution of the material in and on the FCC particles and granulate structure. To meet the criteria for step (i), simple reduction of pressure using a rotovap sufficed. Meeting an optimum for (ii), the distribution of material, is much more complex, and indeed depends strongly on the dynamic of permeation and absorption of the amoxicillin solution into the porous medium, and the time progression for partial dissolution and precipitation/deposition within it. We agree with the Reviewer that further work would be needed to optimise the spatial distribution of the active within the porous medium. We now emphasise this limitation to the work in the text and suggest the need for further optimisation.

5. How, XRD, To understand the underlying mechanisms influencing amoxicillin release, give this understanding?

The Reviewer identifies an important link here between the XRD structure-related properties of the active agent, in this case amoxicillin, and the subsequent desired release into aqueous medium. The Reviewer will recognise the difficulty we faced, and discussed in some depth, regarding the likely structural nature of the active as deposited within the porous FCC particles. We had hoped that the evidence of molecular arrangement provided by XRD would be more convincingly displayed. Given the ambiguity the XRD data raised, we chose to be somewhat reserved in providing a detailed mechanism for the release. We agree this leaves some open questions, and resort to the experience shown by the nutraceutical and flavourings exercise reported earlier by Lundin-Johnson et al., especially noting the similar rapid early stage dissolution occurring during release from FCC in the absence of ethyl cellulose. We reference this difficulty in the manuscript and rely on future work revealing more clearly the specific mechanistic stages involved. Suffice it to say at this point that the controlled dynamic required for the proposed application has been reached despite the lack of a complete molecular-based mechanistic description. We trust this applications-focused result is of sufficient worth to support publication.

6. How author is sure that during casting in organic solvent, microparticle not dissolve/break?

Here we assume that the Reviewer is referring to the stability of the FCC structures. Given the inorganic structural nature of the FCC, we would not expect any microparticle dissolution or breakage. Furthermore, the work reported by

D. Preisig, et al., ‘Drug Loading into Porous Calcium Carbonate Microparticles by Solvent Evaporation,’ Eur. J. Pharm. Biopharm. 87, 548–558 (2014).

T. Stirnimann, et al., ‘Compaction of Functionalized Calcium Carbonate: A Porous and Crystalline Microparticulate Material with a Lamellar Surface,’ Int. J. Pharm. 466(1–2), 266–275 (2014).

T. Stirnimann, et al., ‘Functionalized Calcium Carbonate as a Novel Pharmaceutical Excipient for the Preparation of Orally Dispersible Tablets,’ Pharm. Res. 30(7), 1915–1925 (2013).

and now cited in the references, confirms the stability of the physical structures. The granulation is formed by the compression in close proximity of particle-particle adjacent surface platelets displaying van der Waals attraction, which, without inorganic breakdown, remains ubiquitous throughout any solvent processing. Only mechanical forces would be expected to disturb further the surface platelet structure.

7. Why HPMC K4m and HPMC K4M and Eudragit RS 100 selected as film former?

Here we assume that the Reviewer is referring to use ethyl cellulose (EC) as the film-forming polymer. We would like to clarify that HPMC K4M and Eudragit RS 100 were not part of our formulation work. The selection of EC was based on its well-established use in sustained-release systems, as it is a non-biodegradable yet biocompatible cellulose derivative widely studied for pharmaceutical encapsulation applications. EC is a biodurable polymer, and its molecular structure can be tuned through controlled degrees of substitution—specifically, the number of etherified hydroxyl groups per glucose unit—to achieve tailored solubility and viscosity profiles in various solvents. Moreover, although EC is not rapidly biodegradable under typical physiological conditions, cellulose and its derivatives are considered environmentally friendly materials, as they are ultimately degradable in natural ecosystems through microbial enzymatic activity (e.g., cellulases). This point is now explained in the revised manuscript.

We hope this clarifies the rationale for choosing EC as a film forming polymer.

8. The characterization of film like, Muco adhesion, tensile strength, swelling etc. should be compiled.

Here the Reviewer takes the work forward into the application environment in respect to both mechanical properties and properties in-vivo. We can but agree wholeheartedly that strength, swelling properties and surface interaction of the ethyl cellulose construct with surrounding mucous membranes is of crucial importance. Simply said, we have not moved on to the important applications environment at this stage and such questions are left open. We now draw attention to this next analytical step in the manuscript.

Reviewer #3:

Comment 1: Keep the term fixed eith

---

## [Decision Letter · Decision Letter 1]

21 Aug 2025

Dear Dr. Niga,

Thank you for submitting your manuscript to PLOS ONE. After careful consideration, we feel that it has merit but does not fully meet PLOS ONE’s publication criteria as it currently stands. Therefore, we invite you to submit a revised version of the manuscript that addresses the points raised during the review process.

We look forward to receiving your revised manuscript.<gwmw style="display:none;"></gwmw>

Kind regards,

Mehnath Sivaraj

Academic Editor

PLOS ONE

Journal Requirements:

Reviewers' comments:

Reviewer's Responses to Questions

**Comments to the Author**

Reviewer #1: (No Response)

Reviewer #2: (No Response)

Reviewer #3: All comments have been addressed

Reviewer #4: (No Response)

Reviewer #5: (No Response)

Reviewer #6: (No Response)

2. Is the manuscript technically sound, and do the data support the conclusions?

Reviewer #1: Partly

Reviewer #2: Yes

Reviewer #3: Yes

Reviewer #4: Partly

Reviewer #5: No

Reviewer #6: Yes

3. Has the statistical analysis been performed appropriately and rigorously?

Reviewer #1: Yes

Reviewer #2: I Don't Know

Reviewer #3: Yes

Reviewer #4: No

Reviewer #5: No

Reviewer #6: Yes

4. Have the authors made all data underlying the findings in their manuscript fully available?

Reviewer #1: No

Reviewer #2: Yes

Reviewer #3: Yes

Reviewer #4: No

Reviewer #5: No

Reviewer #6: Yes

5. Is the manuscript presented in an intelligible fashion and written in standard English?

Reviewer #1: No

Reviewer #2: Yes

Reviewer #3: Yes

Reviewer #4: No

Reviewer #5: No

Reviewer #6: Yes

Reviewer #1: Dear Authors,

Thank you for the revisions made according to the reviewer's comments. Some revision has been done well. However please fulfill these following comments;

1. In Fig 4. please give information on the amoxicillin loading for loaded FCC granules as shown in blue squares.

2. If possibly, please provide the UV spectra of FCC fines, FCC granules and, Omyapharm particles (used to prepared EC film, in a range that cover wavelength of 272 nm as Data Availability.

3. Please workout more on the authors' response to reviewer's comment 3 on the aspect of molecular structure of the precipitated amoxicillin and intermolecular structure. In addition, the image of shown in Fig 7 (C) does not help to make relationship between the sustained release of precipitated amoxicillin from FCC. Not only the molecular and intermolecular structure affects the releasement performance but also the physical characteristics of the precipitated of amoxicillin inside the FCC and on the surface.

Reviewer #2: The author has successfully addressed all the comments and suggestions provided in the previous review. The revisions and editions made are satisfactory and have enhanced the overall quality of the manuscript.

Reviewer #3: The paper can be accepted. The comments have been addressed. The author has responded point to point comments

Reviewer #4: This study investigates FCC microparticles integrated in ethyl cellulose films for sustained amoxicillin delivery—an innovative and clinically relevant concept targeting catheter-related infections. The combined use of SEM, UV‑vis, synchrotron XRD, and TGA strengthens the material analysis. Nonetheless, the manuscript currently falls short in methodological completeness, clarity, and novelty justification

Major Issues (Essential for Revision):

1. Material transitions and clarity

o The manuscript frequently switches between FCC fines, granules, and Omyapharm (e.g., in Chapters 2 and 3). A comparative table listing surface area (BET), morphology, rationale, and selection criteria for each form would greatly improve clarity.

2. Film characterization for application

o Films designed for indwelling use require mechanical (tensile strength), swelling, and mucoadhesion data. At minimum, quantitative swelling or tensile information should be included, or limitations acknowledged.

3. Drug release quantification

o Relying on TGA for drug release estimation is unconventional. While UV vis release curves are present, detailed release kinetics (e.g., cumulative release, release rate ± SD, model fitting) must be shown. Clarify that TGA data are supplementary and residual only.

4. Novelty and comparative context

o Similar systems (e.g., Mg HA, Eudragit, MCM 41) have been used for amoxicillin delivery. The manuscript should explicitly highlight what is novel—such as use of FCC in EC films—and compare performance metrics (release duration, loading capacity) to existing systems.

5. Terminology consistency

o Use “sustained” exclusively throughout. Currently, the manuscript alternates between “sustained” and “extended.” Ensure consistent usage.

Minor Suggestions:

• Correct remaining typographical and formatting issues (e.g., “Formatted: Font: 16 pt”).

• Expand figure legends (e.g., labeling apparatus components in Figure 2).

• Define all solvents and abbreviations in the Methods section first mention.

• Clarify that ethyl cellulose and FCC are biocompatible and sterile, as relevant to medical application.

The study is innovative and scientifically sound, but essential methodological work and clarity improvements are necessary for full publication. I encourage the authors to address these points comprehensively.

Reviewer #5: I recommend that the authors substantially revise the manuscript to include detailed methodologies, robust characterization data, well-annotated images, and thorough discussions. Doing so will significantly improve the clarity, reproducibility, and scientific impact of the work.

Reviewer #6: I can recommend publishing, however some the authors need to answer the following question and accordingly revise the manuscript.

1) How did the authors determine crystal unit cell parameters from a bench-top XRAY machine. Did they index the powder pattern ? If so the results should be documented ? If they grew single crystal , then did the simulated pattern from single crystal match the experimental pattern ? Has the structure been uploaded to the CCDC database ?. These details are missing and important to understand how these results were obtained.

2) For the different dissolution experiments the authors have not described whether the tests were conducted under sink or non-sink conditions ? If under sink conditions, how will this correlate to drug release from catheders once implanted ?

3) Did the authors do a mass balance experiment and confirm that at the end of 5 month dissolution experiments the amoxicillin loaded in omyapharm had the remaining drug in the intact in the films or had the drug degraded by that time ?

4) There is a lack of statistical data in the dissolution release models? No error bars seen and std deviation mentioned in the experiments.

.

Reviewer #1: No

Reviewer #2: No

Reviewer #3: No

Reviewer #4: No

Reviewer #5: No

Reviewer #6: No

---

## [Author Response · Author response to Decision Letter 2]

24 Oct 2025

The response to reviewers is loaded as a separate file.

---

## [Decision Letter · Decision Letter 2]

26 Nov 2025

Dear Dr. Niga,

Thank you for submitting your manuscript to PLOS ONE. After careful consideration, we feel that it has merit but does not fully meet PLOS ONE’s publication criteria as it currently stands. Therefore, we invite you to submit a revised version of the manuscript that addresses the points raised during the review process.

We look forward to receiving your revised manuscript.

Kind regards,

Mehnath Sivaraj

Academic Editor

PLOS ONE

Journal Requirements:

Reviewers' comments:

Reviewer's Responses to Questions

**Comments to the Author**

Reviewer #1: All comments have been addressed

Reviewer #4: (No Response)

Reviewer #5: All comments have been addressed

Reviewer #6: All comments have been addressed

2. Is the manuscript technically sound, and do the data support the conclusions?

Reviewer #1: (No Response)

Reviewer #4: Partly

Reviewer #5: Yes

Reviewer #6: Yes

3. Has the statistical analysis been performed appropriately and rigorously?

Reviewer #1: (No Response)

Reviewer #4: Yes

Reviewer #5: No

Reviewer #6: N/A

4. Have the authors made all data underlying the findings in their manuscript fully available?

Reviewer #1: (No Response)

Reviewer #4: Yes

Reviewer #5: Yes

Reviewer #6: Yes

5. Is the manuscript presented in an intelligible fashion and written in standard English?

Reviewer #1: (No Response)

Reviewer #4: Yes

Reviewer #5: Yes

Reviewer #6: Yes

Reviewer #1: Dear Author,

The revisions are fully fulfilled to all points of previous comments and suggestions.

Reviewer #4: Thank you for the thoughtful revision. The manuscript is much improved in clarity and positioning. I appreciate the addition of the FCC materials comparison (particle size/BET/bulk density with usage rationale), the clearer description of sink (particles) vs non-sink (films) conditions, the inclusion of UV-vis controls showing no FCC absorbance at 272 nm, and the expanded comparison to prior amoxicillin delivery systems. The sustained multi-month release under non-sink conditions is a useful proof-of-concept for indwelling applications.

Recommendation: Minor Revision

What has been satisfactorily addressed

Materials clarity: Table summarizing fines, granules, and Omyapharm with BET/density and rationale greatly improves readability.

Terminology: Consistent, correct use of “sustained release” with a brief definition.

Context/novelty: Stronger discussion vs MCM-41, Mg-HA, Eudragit, hydrogels, hollow silica; the FCC-in-EC approach and months-long profile are now well framed.

Analytical specificity: UV spectra for FCC materials confirm no absorbance at 272 nm, supporting the release measurements.

Requests before acceptance (minor but important)

Statistics & replication for release data.

Please report the number of independent replicates for particle (Figs 3–4) and film (Figs 8–9) release experiments. Where feasible, provide mean ± SD and indicate n in figure captions. If the 5-month film curve was run on a single representative sample (as stated), keep it as proof-of-concept but clearly label it as such in the text and captions.

Basic kinetic analysis.

Add simple model fits (e.g., Higuchi and/or Korsmeyer–Peppas) to the film data—separating the initial burst and the longer sustained phase is fine. A small SI table with parameters (and R², n for Korsmeyer–Peppas) is sufficient. If you believe non-sink conditions make a specific model inapplicable, a brief rationale will suffice.

End-point mass balance for films.

Provide an assay at the end of the 5-month study confirming remaining drug in the films and commenting on degradation. TGA (as used for particles) is acceptable; HPLC/LC-MS would be ideal if available. This will strengthen the long-duration result.

Compact methods summary.

Include a short table (main text or SI) listing, for each experiment: matrix (particles/films), sink vs non-sink, total volume, temperature, agitation/flow, exposed surface area (your coiled film geometry is clear—just recap the calculated area). This will improve reproducibility.

XRD reporting clarity.

Please clarify whether the powder pattern was indexed and temper any structural claims that would require single-crystal data. A one-sentence note is enough; a CCDC deposition is not necessary here.

Limitations statement (applications).

Keep/strengthen the explicit note that tensile strength, swelling, and mucoadhesion of EC films were not examined in this proof-of-concept and are planned for future work toward indwelling use.

Minor edits

Remove any residual tracked-changes artifacts (e.g., “Formatted: …”).

Keep explicit loading information in Fig. 4 caption.

If you add statistics/kinetic fits/mass-balance data to SI, please reference them in the Data Availability statement and relevant figure captions.

With the above clarifications and small additions, the manuscript will meet PLOS ONE’s rigor and transparency standards.

Reviewer #5: (No Response)

Reviewer #6: I support publication, but I would like the authors to clarify the following before acceptance.

1) We can speculate that the driving mechanism for nanocrystal formation could be related to a sorption effect of molecules in contact with a surface which could provide a nanoscale molecular orientation effect, a conformation that prevents the lowest energy intermolecular structure, namely preventing crystal formation.

• Could the authors clarify "lowest energy intermolecular structure"? Amorphous amoxicillin suggests a high-energy metastable state, while the crystalline form represents a low-energy conformation.

2) Did the authors examine, or can they propose, whether amoxicillin underwent crystallization during the release studies? This consideration could potentially account for the observed release profile in relation to the differences in solubility between crystalline and amorphous forms of amoxicillin.

3) How does the solubility of amorphous amoxicillin compare to crystalline amoxicillin? How can changes in crystal form during formulation affect the product's performance and release?

.

Reviewer #1: No

Reviewer #4: No

Reviewer #5: No

Reviewer #6: No

---

## [Author Response · Author response to Decision Letter 3]

5 Dec 2025

The Response to Reviewers is attached as a separate document.

---

## [Decision Letter · Decision Letter 3]

19 Jan 2026

Dear Dr. Niga,

Thank you for submitting your manuscript to PLOS ONE. After careful consideration, we feel that it has merit but does not fully meet PLOS ONE’s publication criteria as it currently stands. Therefore, we invite you to submit a revised version of the manuscript that addresses the points raised during the review process.

We look forward to receiving your revised manuscript.

Kind regards,

Mehnath Sivaraj

Academic Editor

PLOS One

Journal Requirements:

Reviewer's Responses to Questions

**Comments to the Author**

Reviewer #4: All comments have been addressed

Reviewer #6: All comments have been addressed

2. Is the manuscript technically sound, and do the data support the conclusions?

Reviewer #4: Yes

Reviewer #6: Yes

3. Has the statistical analysis been performed appropriately and rigorously?

Reviewer #4: N/A

Reviewer #6: Yes

4. Have the authors made all data underlying the findings in their manuscript fully available?

Reviewer #4: Yes

Reviewer #6: Yes

5. Is the manuscript presented in an intelligible fashion and written in standard English?

Reviewer #4: Yes

Reviewer #6: Yes

Reviewer #4: The authors have carefully and satisfactorily addressed the major points raised in the previous review. The revised manuscript is substantially clearer, particularly regarding the distinction and rationale for the different FCC materials, the positioning of TGA versus UV–vis measurements for drug release, and the discussion of novelty relative to existing amoxicillin delivery systems. Terminology is now consistent, and the scope of the work is appropriately framed as a proof-of-concept study focused on sustained release behavior.

While mechanical and swelling properties of the films and statistical replication of long-term release experiments remain outside the scope of the present work, these limitations are now explicitly acknowledged and no longer overstated in the conclusions. The experimental data presented support the claims made within this defined scope.

Only minor points remain, mainly editorial in nature (e.g., wording, figure labeling, and formatting). Detailed comments and suggestions are provided in the attached review letter. These points do not affect the scientific validity of the study.

Overall, the manuscript is technically sound and suitable for publication after minor editorial adjustments.

Reviewer #6: All comments have been addressed satisfactorily. No further questions. I can now recommend publishing.

.

Reviewer #4: **Yes:** Sepideh AsadiSepideh AsadiSepideh AsadiSepideh Asadi

Reviewer #6: No

---

## [Author Response · Author response to Decision Letter 4]

10 Feb 2026

6. Review Comments to the Author

Reviewer #4: The authors have carefully and satisfactorily addressed the major points raised in the previous review. The revised manuscript is substantially clearer, particularly regarding the distinction and rationale for the different FCC materials, the positioning of TGA versus UV–vis measurements for drug release, and the discussion of novelty relative to existing amoxicillin delivery systems. Terminology is now consistent, and the scope of the work is appropriately framed as a proof-of-concept study focused on sustained release behavior.

While mechanical and swelling properties of the films and statistical replication of long-term release experiments remain outside the scope of the present work, these limitations are now explicitly acknowledged and no longer overstated in the conclusions. The experimental data presented support the claims made within this defined scope.

Only minor points remain, mainly editorial in nature (e.g., wording, figure labeling, and formatting).

Detailed comments and suggestions are provided in the attached review letter. These points do not affect the scientific validity of the study.

Overall, the manuscript is technically sound and suitable for publication after minor editorial adjustments.

Remaining Issues and Minor Comments

1. Statistical and quantitative language

Long-term release experiments are still based on single representative samples. While

this limitation is acknowledged, language such as “average release rate” should be

softened to avoid over-interpretation. Terms such as “apparent” or “estimated” release

rate would be more appropriate in this context.

2. Figure clarity

Figures 8 and 9 would benefit from clearer, more explicit labeling of “loaded” versus

“physical mixture” directly on the plots to improve readability without requiring

extensive reference to the captions or text.

3. Formatting and presentation

A small number of formatting artifacts and spacing inconsistencies remain in the revised

manuscript and should be corrected during final editing.

4. MIC and biological relevance

The discussion linking sustained release behavior to MIC values is informative; however,

it should remain cautiously framed as indicative only. Care should be taken to avoid

implying predictive relevance for in vivo efficacy at this stage.

These points do not affect the scientific validity of the study.

Overall, the manuscript is technically sound and suitable for publication after minor editorial adjustments.

Response to reviewer #4

We thank the reviewer for these constructive and helpful final comments, which have improved the clarity of the manuscript.

Remaining Issues and Minor Comments

1. Statistical and quantitative language

We agree and have revised the manuscript to replace “average release rate” with “estimated release rate” to reflect that the long-term release data are based on single representative samples.

2. Figure clarity

We have retained the legend and added explicit curve labels directly within Figures 8 and 9 to clearly distinguish “loaded” and “physical mixture” samples.

3. Formatting and presentation

We have carefully reviewed the manuscript and corrected remaining formatting artifacts and spacing inconsistencies during final editing.

4. MIC and biological relevance

We agree and have revised the MIC discussion to ensure it is framed as indicative only and does not imply predictive in vivo efficacy.

Reviewer #6: All comments have been addressed satisfactorily. No further questions. I can now recommend publishing.

Response to Reviewer #6

We thank the reviewer for their careful evaluation and for confirming that all comments have been satisfactorily addressed.

---

## [Decision Letter · Decision Letter 4]

10 Mar 2026

Functionalized Calcium Carbonate Microparticles in Ethyl Cellulose Films: A Vehicle for Sustained Amoxicillin Release for Medical Applications

PONE-D-25-08617R4

Dear Dr. Niga,

We’re pleased to inform you that your manuscript has been judged scientifically suitable for publication and will be formally accepted for publication once it meets all outstanding technical requirements.

Kind regards,

Mehnath Sivaraj

Academic Editor

PLOS One

Additional Editor Comments (optional):

Reviewers' comments:

Reviewer's Responses to Questions

**Comments to the Author**

Reviewer #3: All comments have been addressed

Reviewer #4: All comments have been addressed

2. Is the manuscript technically sound, and do the data support the conclusions?

Reviewer #3: Yes

Reviewer #4: Yes

3. Has the statistical analysis been performed appropriately and rigorously?

Reviewer #3: Yes

Reviewer #4: Yes

4. Have the authors made all data underlying the findings in their manuscript fully available?

Reviewer #3: Yes

Reviewer #4: Yes

5. Is the manuscript presented in an intelligible fashion and written in standard English?

Reviewer #3: Yes

Reviewer #4: Yes

Reviewer #3: This paper has addressed all the comments. This paper can be accepted now. Please arrange the references according to guidelines.

Reviewer #4: (No Response)

.

Reviewer #3: No

Reviewer #4: No

---

## [Editor Report · Acceptance letter]

PONE-D-25-08617R4

PLOS One

Dear Dr. Niga,

I'm pleased to inform you that your manuscript has been deemed suitable for publication in PLOS One. Congratulations! Your manuscript is now being handed over to our production team.

Kind regards,

on behalf of

Dr. Mehnath Sivaraj

Academic Editor

PLOS One